# Eukaryote Genes Are More Likely than Prokaryote Genes to Be Composites

**DOI:** 10.3390/genes10090648

**Published:** 2019-08-28

**Authors:** Yaqing Ou, James O. McInerney

**Affiliations:** 1Division of Evolution and Genomic Sciences, School of Biological Sciences, Faculty of Biology, Medicine and Health, The University of Manchester, Manchester M13 9PL, UK; 2School of Life Sciences, University of Nottingham, Nottingham NG7 2UH, UK

**Keywords:** composite genes, sequence similarity networks, odds ratio test

## Abstract

The formation of new genes by combining parts of existing genes is an important evolutionary process. Remodelled genes, which we call composites, have been investigated in many species, however, their distribution across all of life is still unknown. We set out to examine the extent to which genomes from cells and mobile genetic elements contain composite genes. We identify composite genes as those that show partial homology to at least two unrelated component genes. In order to identify composite and component genes, we constructed sequence similarity networks (SSNs) of more than one million genes from all three domains of life, as well as viruses and plasmids. We identified non-transitive triplets of nodes in this network and explored the homology relationships in these triplets to see if the middle nodes were indeed composite genes. In total, we identified 221,043 (18.57%) composites genes, which were distributed across all genomic and functional categories. In particular, the presence of composite genes is statistically more likely in eukaryotes than prokaryotes.

## 1. Introduction

Reticulation occurs when two or more evolutionary lineages merge, and consequently, reticulation cannot be visualised or analysed using tree-like models of evolution. We see reticulate events occurring during meiotic recombination, horizontal gene transfer (HGT, also known as lateral gene transfer) [1], exon shuffling [2], and hybrid speciation [3] for example. Merger events can be seen at multiple levels, such as genes, genomes, operons and gene clusters.

This paper focuses on the combination of genetic fragments from unrelated gene families to produce a single gene. This process of gene fusion occurs when parental (or component) genes merge to form a new gene called a composite (or fused) gene [2,4]. Because reticulate evolution cannot be adequately represented using tree-like representations, we constructed sequence similarity networks (SSNs, also known as protein/gene similarity networks) and visualised them using Gephi [5] and Cytoscape [6]. In these kinds of networks, gene, genome or species data can be used to detect recombination events. In the SSNs that we have constructed, genes or proteins are represented as nodes while inferences of homology between genes are represented by edges. Within the framework of the SSN, some special relationships, such as non-transitive triplets when two component genes have no overlap, can be identified as motifs in the network. SSNs have been used elsewhere in order to investigate the existence of composite genes [7,8]. In an analysis of 15 eukaryotic genomes Haggerty et al. [7] constructed a network that contained a giant connected component (GCC) where one quarter of all sequences were identified as composite genes and approximately 10% of sequences were identified as multi-composite genes (those formed from the union of two or more composite genes). Moreover, Coleman et al. [8] used SSNs to explore 1642 antibiotic resistance genes derived from more than 100 species. They found 73 fused genes using the FusedTriplets software [9,10], which accounted for 4.43% of the total gene count. In addition, Jachiet et al. [10], using the MosaicFinder software, found gene fusions in both cellular organisms and mobile genetic elements (MGEs). In another analysis using the same kind of approach, viruses were suggested to consist of only 8–15% of composite genes, with this low number being attributed to the blurry boundaries between viral gene families [11]. In addition, gene fusion has been shown to have played an essential role in the evolution of the cellular life cycle, with composite gene formation seen in genes related to chromatin structure and nucleotide metabolism [12]. Also, Ocaña-Pallarès et al. [13] concluded that there was a significant role for gene fusion in the origin of eukaryotes, as evidenced by SSN built from eukaryotic EUKaryotic restricted Nitrate Reductase (EUKNR) and similar eukaryotic and prokaryotic sequences. The result indicated that EUKNR was formed by a fusion of eukaryotic sulfite oxidases (SUOX, N-terminal) and NADH (C-terminal) reductases. Therefore, while it is clear that gene fusion is a common feature of genes, a comprehensive comparison across a broad range of taxa and molecule types would provide more evidence for its frequency and impact.

In this paper, we describe an approach to identify composite genes using a dataset of 1875 completed genomes, comprising more than one million sequences, from all three domains of life as well as from MGEs. We tested whether the rate of gene remodelling has been uniform across all of life, and all cellular functional categories.

## 2. Materials and Methods 

### 2.1. Dataset Construction and BLAST Analysis

A total of 1,190,265 protein sequences were collected from the RefSeq database at the National Centre for Biotechnology Information [14]. We manually selected taxa in order to maximise diversity, while also ensuring computational tractability. The final dataset covered 261 species from the main representative lineages, belonging to 36 eukaryotes (13 phyla, 21 classes), 56 archaea (4 phyla, 9 classes), 90 bacteria (25 phyla, 32 classes), 79 viruses and 1,614 plasmids. Homology between pairs of amino acid sequences was inferred using an all-versus-all protein BLAST (BLASTP version 2.4.0, NCBI, Bethesda, MD, United States), with an E-value cutoff of 1e-5, 5000 max target sequences, and soft masking parameter (the others by default) [15]. The dataset species information and download paths are available at https://github.com/JMcInerneyLab/CompositeGenes/blob/master/accession.txt.

### 2.2. Composite Gene Identification

Using the BLAST results as input, we identified composite genes as motifs of triplets in the graph where there was a “non-transitive” relationship between three nodes [4]. Composite gene detection was carried out by the CompositeSearch program [16] when associated component genes have no overlap theoretically, with default identity cutoff of 30% and 20 amino acid overlaps to limit false negative error. The CompositeSearch output contains information on composite genes, component genes and the families to which they belong. This output was depicted, explored, and manipulated using Gephi (version 0.9.2, The Gephi Consortium, Paris, France) [5]. 

Because the proportion of composite gene from different domains might be affected by biased sequence database sampling, we randomly sampled 50,000 genes from archaea, bacteria, eukaryotes and plasmids respectively. These random samples were taken forward for analysis in the same way as the original data. The major difference between the sub-sampled datasets and the original data was that in the subsampled datasets, the number of genes from each of the four kinds of dataset was the same. We used CompositeSearch in order to construct an SSN from the BLASTP output of the subsampled datasets containing 200,000 genes. These SSNs were then used in order to identify composite genes. Sampling was repeated 100 times and the results were summarised graphically.

### 2.3. Functional Annotations

We used EggNOG (version 4.5.1, Computational Biology Group–EMBL, Heidelberg, Germany) [17] in order to assign gene functional categories. The analysis was carried out through the web interface using the DIAMOND [18] mapping mode. In the output, genes were assigned to different Orthologous Groups (OGs), and each OG had functional annotations that included Clusters of Orthologous Groups (COGs) functional categories: COG for universal Bacteria, EuKaryotic Orthologous Groups (KOGs) for Eukaryotes and arKOGs for Archaea [19]; Gene Ontology (GO) terms [20]; Kyoto Encyclopedia of Genes and Genomes (KEGG) pathways and SMART/Pfam protein domains. Both composite and non-composite genes were placed into at least one of 23 COG categories and at least one of four GO terms.

### 2.4. Statistical Analysis

In the EggNOG output, each gene has a detailed functional annotation and is associated with at least one general COG category code (A to Z apart from R and X). Because of recombination, the category code for a given gene could be single letter like ‘A’ or multiple letters such as ‘ABC’. When counting the number of genes that possess a particular function, if a multiple letter category was selected, we counted this gene multiple times. For instance, if the most common COG category for a gene was ‘ABC’, and then this gene was counted three times as A, B and C.

To investigate the distribution of composite genes and non-composite genes among eukaryotes and prokaryotes, an odds ratio (OR) test [21] was carried out. OR tests are normally used to test the strength of the association between two events. Here, for each protein function, we used an OR test to test the association between gene origin and the likelihood of fusion Equation (1).
(1)$$OR\;=\frac{a/c}{b/d}=\;\frac{ad}{bc}$$
where *a* is the number of composite eukaryote genes, *b* is the number of non-composite eukaryote genes, *c* is the number of composite prokaryote genes, *d* is the number of non-composite prokaryote genes. The 95% confidence intervals (CI) were calculated by
Upper 95% CI=e^[ln(OR)+1.96(1a+1b+1c+1d)]
(2)Lower 95% CI=e^[ln(OR)−1.96(1a+1b+1c+1d)]

For all OR tests that were carried out on 24 COG categories, we used a conservative Bonferroni correction [22] to limit type I error. The critical level of significance was initially set as *α* = 5%, we corrected it as *α*/2*N*, *N* is the number of performed tests, which in our case is 24. The new significance level is 0.1% and corresponding confidence coefficient of 99.9% is 3.09 standard deviations, using the standard normal distribution table. The corrected CI was calculated by
Upper 95% CI=e^[ln(OR)+3.09(1a+1b+1c+1d)]
(3)Lower 95% CI=e^[ln(OR)−3.09(1a+1b+1c+1d)]


## 3. Results

### 3.1. Pervasive Existence of Composite Genes across All of Life

We assembled a dataset of 1,190,256 genes from 36 eukaryotes, 56 archaea, 90 bacteria, 79 viruses and 1614 plasmids from more than 60 taxonomic classes. Following an all-versus-all BLAST, a total of 540,325,758 significant hits were detected. Using CompositeSearch, an SSN containing 1,025,263 nodes and 109,650,422 edges was constructed. In this network, 221,043 composite genes (18.57% of the gene dataset, Figure 1a) were identified, linked to 603,604 component genes. Collectively, these genes were assigned to 360,981 gene families.

To gain a better understanding of those genes involved in non-homologous recombination, all genes were categorized into four groups: nested composite genes, strict composite genes, strict component genes, and non-remodelled genes (Figure 2). Nested composite genes have been formed by the merging of at least two sequences but are additionally involved in other non-transitive triplets as components; that is to say they themselves are composites but also form other composites. In contrast to nested composites, strict composite genes only act as composite genes in the network, similar to strict component genes. Non-remodelled genes do not show evidence of having participated in any recombination events. In our dataset, 181,157 genes as nested composite genes, 39,886 genes were identified as strict composite genes, 422,447 as strict component genes, and 546,775 as non-remodelled genes (Figure 3a).

Within 182 species across the three domains of life, remodelled composite genes were discovered in all species, indicating that gene fusion is, and has been, widespread across all life on Earth. Overall, 23.66% (205,913 composites identified from 870,120 eukaryotic and prokaryotic genes) of examined genes were identified as composite. However, there was a considerable amount of variation in the proportion of composite genes across species and molecule type. Table 1 presents the ten genomes with the highest and lowest rates of composite genes among eukaryotes and prokaryotes. Composite genes account for almost one third of the genomes of *Homo sapiens*, *Volvox carteri f. nagariensis* and *Aureococcus anophagefferens*.

As shown in Figure 1a, the proportion of composite genes often shows a wide distribution, depending on the classification of the genome in which the gene is found. Among cellular lifeforms, eukaryote genomes contain the highest proportion of composite genes on average (22.66%), followed by bacteria (14.76%) and then archaea (12.78%). However, the distributions are quite wide though prokaryote species manifested a narrower distribution of composite frequency when compared with eukaryotes. When considering mobile genetic elements, the average percentage of composite genes in plasmids (14.69%) is almost the same as bacteria but is noticeably higher than the average seen for virus genes (4.82%).

To avoid the effects of unequal sampling in large dataset, we used a jackknife resampling approach in order to generate datasets of 50,000 sequences each from eukaryotes, archaea, bacteria and plasmids. With these uniformly-sized gene sets we used the same analysis methods as for the large dataset: sampling, identifying homologs and constructing SSNs. We then replicated this process 100 times. On average, across all replicates, 19,443 (9.72%) genes were identified as composite genes (Figure 1b), which is approximately half the percentage identified from the large dataset (18.57%). The difference indicates that the detection rate of composite genes is related to genomic sequence sampling size and therefore, the reporting of composite genes is always a lower bound for the actual percentage. The resampling procedure was designed to analyse composite gene distribution while attempting to normalise for the difference in data size for each of the four main classifications (eukaryote, bacteria, archaea and plasmids). Plasmids have the highest proportion of strict composite genes while eukaryotes have the largest proportion of nested composite genes (Figure 1b). Nonetheless, even though there is no obvious difference between eukaryotes and prokaryotes in terms of the number of nested composite genes, strict composite genes are approximately twice as likely in eukaryotes as in archaea and bacteria. Bacteria and archaea are quite similar, in terms of their proportions, for all four categories of remodelled and non-remodelled genes. Finally, strict component genes do not show much difference across any of our genome types though eukaryotes have the highest number of strict components but the lowest number non-remodelled genes.

### 3.2. Sequence Functional Annotations

The EggNOG mapper program [17] was used to assign functions to all sequences. For all results, COG and GO annotations were used to evaluate functional categories. First, composite genes were found to be widespread across all functional categories (Figure 3, Appendix A). Gene distributions show different patterns across different functions (Figure 3a, Appendix A). Genes with unknown function (category S, 66.23% non-remodelled) are less likely to have been remodelled. The category of genes that have the second-lowest rate of remodelling is cell motility (N, 49.08% non-remodelled). Genes in RNA processing and modification (A, 26.52%) and dynamics (B, 25.96%) had the highest rate of nested composites. Conversely, genes involved in signal transduction (T, 59.05%) tend to have the highest proportion of strict component genes whereas genes involved in extracellular structures (W, 7.3%) are more likely to be strict composite.

We used an odds ratio (OR) test and Bonferroni correction (see Methods) on composite and non-composite genes from eukaryotes and prokaryotes in different functional categories in order to understand if genes from different classifications were more likely to be remodelled in one or the other. If the OR value and its upper and lower 95% CI value span 1, we take this as evidence that there is no significant difference in composite gene formation between eukaryotes and prokaryotes, and vice versa. If the OR number is greater than 1, this indicates a positive correlation between remodelling and being from a eukaryotic genome, while if the number is less than 1, it indicates an association between remodelling and being a prokaryote. From the results of these analyses, the frequency of composite genes in eukaryotes were found to be statistically higher than from that of prokaryotes for most kinds of gene (Figure 3b, Appendix A, Appendix A). Some exceptions were found for genes in extracellular structures (category W) and RNA processing (category A) whose 95% CI was found to span 1 (Figure 3b). Therefore, across all the species examined, the odds of a gene being a composite if it is a eukaryote is statistically significant higher than if it is a prokaryote.

## 4. Discussion

Network models such as SSNs have been broadly employed in studies of evolutionary relationships [23] and gene sharing and recombination detection. We carried out a large-scale examination of more than one million genes across 1875 complete proteomes including archaea, bacteria, eukaryote, plasmids and viruses. The results suggest that composite genes exist in all organisms and across all kinds of genes.

Eukaryotes, are known to have originated from the merger of an archaeon and a bacterium [24]. On average, more than one fifth of eukaryote genes show evidence of remodelling by gene fusion and the probability of a gene in our dataset being composite if it is derived from a eukaryote genome are significantly higher than the probability if the genes comes from a prokaryote genome. What is not known at this stage is the process that has led to the change in frequency of gene remodelling. Candidates for the process include the combination of homologous recombination during meiosis, combined with the relatively lower level of horizontal gene transfer (HGT) in eukaryotes compared with prokaryotes. The lower level of HGT means that evolutionary innovation via HGT is more restricted in eukaryotes and this restriction, combined with the opportunities for illegitimate crossover events during meiosis could account for the elevated levels of remodelling. In other words, restricting HGT sets up a situation where composite gene formation is one of the main routes to evolutionary innovation. These findings are consistent with Jachiet et al. [10] who found that eukaryote sequence evolution was highly influenced by gene fusion.

Although evidence of remodelling is quite high in eukaryote genes, plasmid genes also show evidence for a large number of gene fusion events. The average percentage composite genes found in plasmid genomes in our dataset is 14.69%, which is almost as high as the percentage recorded for bacteria. In 2013, Jachiet et al. [10] mined a data set from three domains of life and MGEs, discovered 42% of composite genes were included at least one MGE gene as a component. Likewise, Halary et al. [25] found that the plasmids in Borrelia genes behaved like “private genetic goods” [26] and were much less likely to be involved in gene remodelling or sharing with other taxa. It has been suggested that this restriction in gene sharing contributed to the survival of Borrelia against the host immune environment [27,28]. The high level of remodelling seen in plasmid genes would suggest that MGEs act as a source for remodelling. Corel et al. also found that gene externalization (gene fusion between cellular organism and MGE) played an important role in microbial evolution [29].

In our dataset, compared to non-composite genes, fusion genes are more likely to be involved in chromatin structure and dynamics, extracellular RNA processing and modification, as well as cytoskeleton. It has already been shown for eukaryotes that composite genes have been foundational [12], particularly in photosynthetic lineages (such as ubiquitin-nickel superoxide dismutase fusion protein in algae [30]). Further, a recent published work by McCartney et al. suggested novel functional protein coding genes in human could emerge through transcription-derived gene fusion [31]. Novel composite genes also have been reported in the origin of haloarchaeal lineages contributed by bacteria, which is named as chimeric (ChiC) genes [32]. ChiC genes are more likely to be involved transport and metabolism whereas other composite genes more likely to be involved in replication, recombination and repair, both functions have high composite gene portion in my dataset. In additional, the research from Corel et al. also suggested that recent externalized genes in abundant in replication, recombination, and repair but hard to accumulate, which could be the result of transposon [29]. Moreover, composite genes in viruses tend to be found in nucleotide metabolism and transport, replication and repair, cell wall, membrane and envelope biogenesis as well as post-translational modification. This finding is consistent with Jachiet et al. [11].

## 5. Conclusions

In conclusion, we applied a network approach in order to investigate composite gene in species across all of life, although the results of this study really only provide a lower-bounds estimate of the extent of gene remodelling, we have been able to show that it is a pervasive and important element of evolutionary history.

## Figures and Tables

**Figure 1 genes-10-00648-f001:**
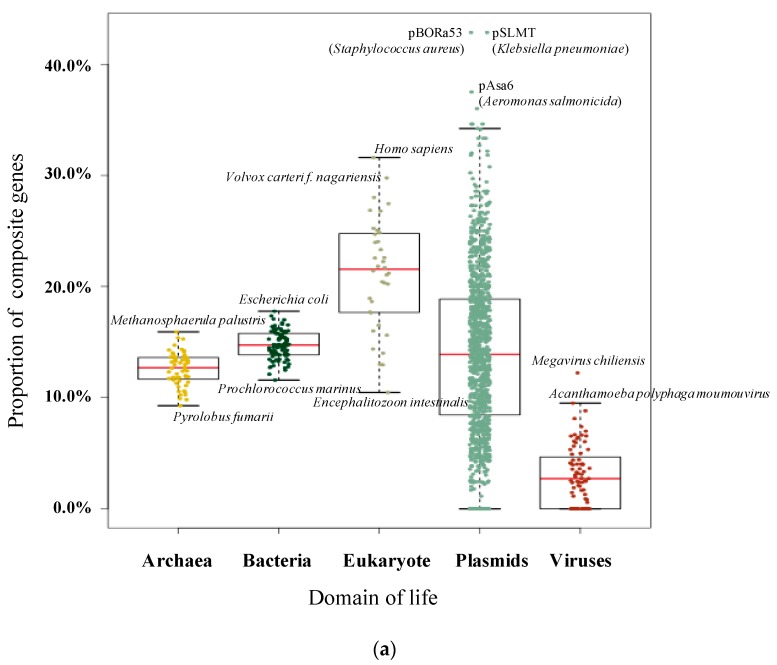
Pictures of composite and component genes among different domains. (**a**) Proportion of composite genes within different domains and mobile genetic elements. Dots represent individual genomes. (**b**) Numbers of nested composite, strict composite, strict component, and non-remodelled genes within different domains for each of the 100 replicates of equal sampling. All analyses were replicated 100 times and each replicate is represented by a dot.

**Figure 2 genes-10-00648-f002:**
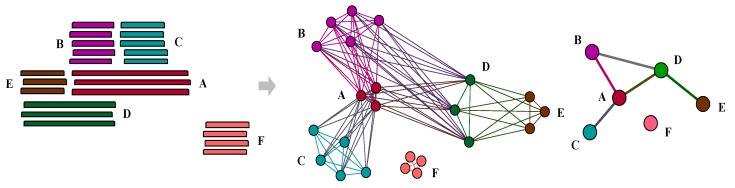
Sample patterns of nested composite, strict composite, strict component, and non-remodelled gene families. Genes in family A not only participate in the fusion of genes in family D as component genes but are also formed by genes from family B and C as composite gens; this is regarded as nested composite genes. In contrast, genes in family B, C and E belong to strict component families which only act as component genes in this network. Similarly, genes in family D as members of a strict composite family. In additional, family F is non-remodelled gene. Also, because there is no overlap between gene family A and C, gene family B and E so ‘A-B-C’ and ‘B-D-E’ can be regarded as non-transitive triplets.

**Figure 3 genes-10-00648-f003:**
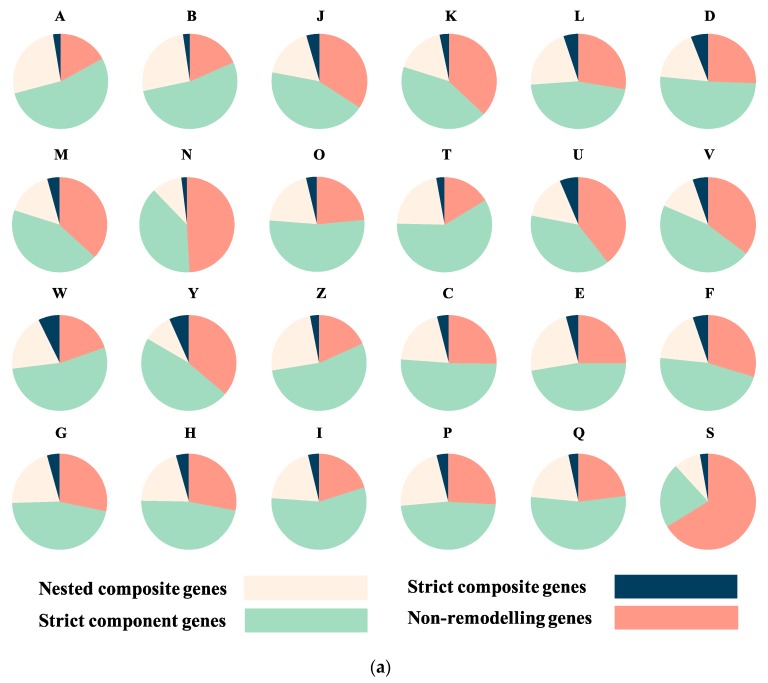
Function analysis of composite and non-composite genes. (**a**) Numbers of nested composite, strict composite, strict component, and non-remodelled genes across all COG categories. (**b**) Numbers of OR, upper 95% CI and lower 95% CI value (after Bonferroni correction) across all COG functions. The detailed numbers are shown in Appendix A. There was not composite gene identified from prokaryote in category Y in this dataset, so OR test was not applied. Apart from A and W, which span 1.0, the odds of composite gene presence in all COG categories shows statistically significant tendency in eukaryotes. A: RNA processing and modification; B: chromatin structure and dynamics; C: energy production and conversion; D: cell cycle control and mitosis; E: amino acid metabolism and transport; F: nucleotide metabolism and transport; G: carbohydrate metabolism and transport; H: coenzyme metabolism; I: lipid metabolism; J: translation; K: transcription; L: replication and repair; M: cell wall/membrane/envelope biogenesis; N: Cell motility; O: post-translational modification: protein turnover, chaperone functions; P: Inorganic ion transport and metabolism; Q: secondary metabolites biosynthesis: transport and catabolism; T: signal transduction; U: intracellular trafficking and secretion; V: defence mechanisms; W: extracellular structures; Y: Nuclear structure; Z: cytoskeleton; S: function unknown.

**Table 1 genes-10-00648-t001:** The ten species that contain the highest (left) and lowest (right) proportions of composite genes. All species that contains more than 24% composite genes are from eukaryotes, whereas most species that contain less than 11% composite genes are from archaea (Crenarchaeota family, mostly).

Species	Total Number of Genes	Number of Composite Genes	Proportion	Species	Total Number of Genes	Number of Composite Genes	Proportion
*Homo sapiens*	109,018	34,455	31.60%	*Fervidicoccus fontis*	1385	152	10.97%
*Volvox carteri* f. nagariensis	14,436	4298	29.77%	*Thermoproteus uzoniensis*	2112	224	10.61%
*Aureococcus anophagefferens*	11,520	3227	28.01%	*Nanoarchaeum equitans*	540	57	10.56%
*Capsaspora owczarzaki*	8792	2413	27.45%	*Staphylothermus marinus*	1598	168	10.51%
*Chlorella variabilis*	9780	2626	26.85%	*Encephalitozoon intestinalis*	1939	203	10.47%
*Polysphondylium pallidum*	12,367	3313	26.79%	*Ignisphaera aggregans*	1930	198	10.26%
*Monosiga brevicollis*	9203	2322	25.23%	*Methanopyrus kandleri*	1687	173	10.25%
*Salpingoeca rosetta*	11,731	2939	25.05%	*Pyrobaculum neutrophilum*	1966	195	9.92%
*Allomyces macrogynus*	19,446	4829	24.83%	*Hyperthermus butylicus*	1681	165	9.82%
*Tetrahymena thermophila*	10,626	2625	24.70%	*Pyrolobus fumarii*	1885	175	9.28%

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
