# Peer review of "Eukaryote Genes Are More Likely than Prokaryote Genes to Be Composites"

_genes, 2019, doi:10.3390/genes10090648_

Round 1

Reviewer 1 Report

In the manuscript by Ou and McInerney, authors described an approach to identify composite genes using a dataset of 1875 complete genome sequences and stated that the presence of composite genes is statistically more likely in eukaryotes than prokaryotes.  Overall, the study is interesting. I have only few comments.

In page 4, 2nd paragraph, authors described 4 groups of genes. I recommend that if possible authors could include the figure in the main text (i.e. as part of fig 1 or fig1). It will be easier for the reader to understand the further details. It is also easier for reader if the authors described in the text in order as stated in the 127-128. The results described the differential proportion of composite genes in different taxa (Fig. 1A) need to discuss scholarly in the discussion section. Is there any possible reason for the discrepancies between mobile genetic elements and other genes? Authors also explored different functional categories (result section 3.2). It is very interesting however; lacks a suitable discussion in the discussion section. I recommend authors should revise the discussion section more scholarly based on the possible biological significance, if any. Sometimes a small but specific genomic portion of a transposable gene could insert into different unrelated genes or insert into some candidate of a multigene family. This affects the gene expression and also function. Is this type of remodeled genes observed in the present study? Minor typos/issues line 29, later>lateral; line 120, delete ‘The text continues here’.

Reviewer 2 Report

Authors identified composites genes by finding non-transitive triplets of nodes in sequence similarity networks. A similar approach was published by the authors in 2014 and 2015. Here, they scale up data set and identify composites genes by an advanced package, CompositeSearch, instead of FusedTriplets. A step forward, the functions of composites genes are annotated through EggNOG. Comparison between eukaryotes and prokaryotes is carefully conducted with the control of sample size effect. The conclusion sounds solid. Few minor concerns are as the following:

The sequence similarity network (SSN) is based on all-versus-all BLASTP whose setting is not clear, i.e., Matrix and Gap Cost. Different setting of BLASTP can affect searching sensitivity and specificity, i.e., BLOSUM80 for insensitive search (Korf I, Yandell M, Bedell J, 2003, BLAST: O’Reilly Media, Incorporated). Further, the stability of SSN should be addressed. For example, SSN in MBE2014 is built by BLAST E-value 1e-10 but here by 1e-5. Less strict E-value BLAST results in more edges of SNN which might affect the proportion of composites gene observed.

line 78 @ page 2
“CompositeSearch program [15], using an identity cut off of 30%.” The explanation of “identity cut off” will be nice. FusedTriplets.py guarantee A and B Blast matches on C do not overlap (MBE2014). I am not sure whether the same criterion is applied in CompositeSearch.

Reviewer 3 Report

Ou & McInerney submission is well-articulated and sound scientific work. It reports a global analysis of composite genes and describes the extent of gene fusions/remodelling across different kingdoms of life. The authors conclude that composites are widespread and significantly more abundant in eukaryotes compared to prokaryotes. Figures, methods and text are clear.

For follow on studies it would be great to see how nucleotide sequence, location and age influence composites.
